# Characterization of an intracellular humanized single-chain antibody to matrix protein (M1) of H5N1 virus

He Sun[1,2], Guangmou Wu[2], Jiyuan Zhang[3], Yu Wang[1,2], Yue Qiu[1,2], Hongyang Man[1,2], Guoli Zhang[2], Zehong Li[1], Yuhuan Yue[2]*, Yuan Tian[2]*

1 College of Life Science, Jilin Agricultural University, Changchun, China, 2 Changchun Veterinary Research Institute, Chinese Academy of Agricultural Sciences, Changchun, China, 3 The Tourism College of Changchun University, Changchun, China

* yhyue2013@163.com (YY); wyaycem@163.com (YT)

**Data Availability Statement:** All relevant data are within the manuscript and its Supporting Information files.

## Abstract

We developed a human intracellular antibody based on the M1 protein from avian influenza virus H5N1 (A/meerkat/Shanghai/SH-1/2012) and then characterized the properties of this antibody. The M1 protein sequence was amplified by RT-PCR using the cDNA of the H5N1 virus as a template, expressed in bacterial expression system BL21 (DE3) and purified. A human strain, high affinity, and single chain antibody (HuScFv) against M1 protein was obtained by phage antibody library screening using M1 as an antigen. A recombinant TAT-HuScFv protein was expressed by fusion with the TAT protein transduction domain (PTD) gene of HIV to prepare a human intracellular antibody against avian influenza virus. Further analysis demonstrated that TAT-HuScFv could inhibit the hemagglutination activity of the 300 $TCID_{50}$ H1N1 virus, thus providing preliminary validation of the universality of the antibody. After two rounds of M1 protein decomposition, the TAT-HuScFv antigen binding site was identified as Alanine (A) at position 239. Collectively, our data describe a recombinant antibody with high binding activity against the conserved sequences of avian influenza viruses. This intracellular recombinant antibody blocked the M1 protein that infected intracellular viruses, thus inhibiting the replication and reproduction of H5N1 viruses.

## Introduction

The H5N1 virus is a highly pathogenic type A avian influenza that can cause systemic or respiratory disease in humans [1]. However, H5N1 is mainly transmitted among birds, and the infection can cause high mortality. The highly pathogenic H5N1 avian influenza virus has infected more than 500 million poultry worldwide since human infection with the H5N1 subtype was first reported in Hong Kong, China, in 1997 [2]. Subsequently, human infections have emerged in Asia, Europe, and Africa, consequently leading to significant public health concerns [3]. According to the antigenic characteristics of the virus and its genetic characteristics, influenza viruses are divided into three types: A, B, and C. Influenza A viruses are divided into many subtypes according to different antigens of hemagglutinin HA and neuraminidase

**Funding:** We received two grants, Jilin Province Science and technology development plan. 20200404113YY, 20190304038YY, Please find the attached file A. The funders had no role in study design, data collection and analysis, decision to publish, or preparation of the manuscript.

**Competing interests:** The authors have declared that no competing interests exist.

NA. HA can be divided into 16 subtypes (H1-H16), and NA has 9 subtypes (N1-N9) [4]. The combination of different subtypes of HA and NA is responsible for the significant diversity of avian influenza viruses.

The influenza virus matrix protein is encoded by viral RNA fragment 7, which contains two open reading frames and therefore can be transcribed into two mRNAs and then translated into M1 and M2 proteins, respectively [5]. The M1 protein, with a molecular weight of approximately 26 kDa, is found extensively in virions, and its sequence is highly conserved. Based on these characteristics, M1 has been used as the basis for classifying influenza viruses into types A, B, and C. M1 protein is expressed in the late stages of viral replication and is concentrated in discrete cell lacunae [6]. A small amount of M1 protein translocates to the nucleus during the late stages of viral infection and helps to inhibit viral transcription. Thereafter, M1 interacts with the NEP (NS2 protein) tail encoded by viral RNA fragment 8 to export vRNP from the nucleus to the cytoplasm [7, 8], thus triggering the budding process and the release of virions. M1 and RNA fragments that encode M1 are the most attractive target sites for antibody drugs, gene silencing, and drug therapy [9, 10].

In this study, a single-chain antibody with high affinity was screened using a phage antibody library, Tomlinson I+J, based on the M1 protein of the H5N1 virus as a target. In view of the transduction properties of protein transduction domain (PTD)-mediated protein across the cell membrane, the PTD of HIV was used to link to the single-chain antibody molecules [11–13]. Following the expression of this segment as a fusion protein with HuScFv, the TAT-HuScFv was efficiently translated into virus-infected cells and bound specifically to the intracellular M1 protein to prevent the assembly and release of the influenza virus. These specific characters of the fusion protein indicate that this humanized single-chain Ab can be used for anti-viral therapy in the future.

## Materials and methods

### Vectors, bacterial strains, helper phage, and libraries

The cloning and expression vector *p*ET-Sumo, *E. coli* strain BL21 (DE3) and SUMO enzyme were purchased from Invitrogen Biotechnology Co. (CA, USA). The *p*ET28a-TAT-GFP vector was constructed previously by our research group [14]. A Tomlinson (I+J) phage antibody library, *E. coli* strains (TG, HB2151), and helper phage (KM13) were obtained from the Medical Research Council in Cambridge, UK. Goat anti-mouse IgG labeled horseradish peroxidase (HRP) antibody and anti-HIS6 labeled antibody were purchased from Abbot Trading Co., Ltd (Shanghai, China). The Protein A-HPR antibody was purchased from Abcam Co. (Cambridge, England). SP Sepharose 4 FF, Chelated Sepharose 4 FF, rProtein-A Sepharose 4 FF, and AKTA Prime chromatographs, were purchased from GE Healthcare (Fairfield, CT, USA). Stocks of H5N1 (A/Meerkat/Shanghai/SH-1/2012) and H1N1 (A/Changchun/01/2009) viruses are held in ourinstitute.

### Expression and purification of H5N1-M1 protein

The cDNA sequence of the M1 protein for H5N1 virus was identified in the NCBI library; this sequence was then used as a template for primer design: forward: 5′–ATG AGT CTT CTA ACC GAG GTC–3′ and reverse: 5′–CCG GAA TTC TTA CTT GAA TCG CTG CAT CTG CAC T–3′. The M1 gene was amplified using H5N1 (A/Meerkat/Shanghai/SH-1/2012) cDNA as a template. The PCR reaction conditions were as follows: 94˚C denaturation for 5 min, 94˚C for 50 s, 50˚C for 50 s, 72˚C for 50 s (for 34 cycles) followed by 72˚C 10 min. The amplification products were separated by gel electrophoresis and then ligated into the PET-SUMO vector with T4 ligase. The vector was sequenced and then transformed into BL21 (DE3) competent cells.

After IPTG induced thallus expression and ultrasonic lysis, the supernatant was taken and precipitated with 20%-45% ammonium sulfate. After centrifugation, the protein was precipitated in 20 mM PB buffer solutions (pH 7.0) and then purified by SP Sepharose 4 FF chromatography. Linear elution of PB containing 0.5 M NaCl was then performed to collect the target protein. Nest, $Cu^{2+}$ metal chelation chromatography was performed, with imidazole at 50 mM eluting the hybrid protein and imidazole at 150 mM eluting the target protein. The target protein was collected and diluted with 20 mM PB buffers (pH 7.0) to an imidazole concentration of 50 mM. SUMO protease was added and digested for 2 h at 30˚C. The product was then digested by $Cu^{2+}$ metal chelation chromatography, and the absorption peak transmission solution without specific binding was collected. The product was then concentrated with SP-FF to obtain the purified M1 protein.

## Biopanning of a phage display library and selection by M1 specificity

The phage antibody library was prepared in accordance with the manufacturer's instructions and then screened with the purified M1 protein as an antigen. During this process, the M1 protein was coated in 96-well plates (Nunc-Nalgene, USA) at a concentration of 5 μg/ well and then incubated at 4˚C overnight. The next day, the supernatant was discarded and washed three times with PBS to remove the non-adsorbed antigen. Non-specific binding was blocked with 200 μL of 2% milk/PBS at 37˚C for 2 h. After discarding the sealing solution, the wells were washed three times with PBS, and the liquid was removed by vigorous shaking. Nest, we added Tomlison I+J phage antibody library (100 μL/well) at a titer of $1.0 \times 10^{13}$ and incubated for 60 min at room temperature with vigorous shaking. After standing for 60 min, the liquid was discarded, and the wells were washed 10 times with PBS (0.05% (V/V) Tween-20). The residual liquid in each well was patted dry and 50μL of eluting solution (5 mg/mL trypsin-PBS) was added to each well. The plates were then shaken at room temperature for 15 min to eluate the phages, which were then stored at 4˚C. The eluted phage was cloned into *E. coli* TG1 and further panning was performed. The second, third, and fourth rounds of panning were performed under similar conditions, except that the concentration of the antigen coating was reduced to 2 μg/well. Unbound phages were removed by 20, 30, and 40 washes with PBS (0.05% (V/V) Tween-20).

Next, 2% milk/PBS (100μL/well) was added to the plate, and the plate was kept overnight at 4˚. Nonspecific binding was then blocked with 2% BSA/PBS for 2 h and the phage antibody from the fourth round of screening was added. After incubation at room temperature for 1 h, the supernatant was collected to remove the phages that had been specifically adsorbed to the milk powder in the antibody library; the collected phages were then stored at 4˚C.

## Expression of positive clones and ELISA analysis for M1 protein

After four rounds of screening, 10 μL of phages were added to 200 μL of fresh *E. coli* HB 2151 and left for 30 min in a water bath at 37˚C. Then, 50 μL was applied to a TYE (15 g bacto-agar, 8 g NaCl, 10 g tryptone, 100 g ampicillin, 10 g glucose, 5 g yeast extract in 1 L) plate and cultured overnight at 37˚C. Once grown on the plate, single colonies were randomly selected and placed on a 96-well culture plate; each well contained 100 μL of 2×TY (30 g bacto-agar, 16 g Nacl, 20 g tryptone, 100 g ampicillin, 10 g glucose, 10 g yeast extract in 1 L) medium and cultured overnight at 37˚C. The next day, approximately 2 μL of bacterial solution from each well was placed in another 96-well cell plate (the remaining solution was added to glycerol at a final concentration of 15% and stored at -70˚C). The new cell plate contained 200 μL of 2×TY medium (containing 100 μg/mL Amp and 0.1% glucose) from each well and cultured at 37˚C to an $OD_{600}$ of 0.9 (after approximately 4 h of culture). Isopropyl β-D-Thiogalactoside (IPTG)

at a final concentration of 1 mmol/L was added to each well and cultured overnight on a 30°C shaker. After overnight culture, the bacterial solution was centrifuged at $1800 \times g$ for 15 min; the supernatant was then transferred to a new plate and stored at 4°C to await testing.

M1 protein (2 μg, 100 μL/well) was added to 96-well plates and incubated overnight at 4°C. The next day, the plates were washed three times (3 min each time) with wash solution (0.05% PBS (V/V) and Tween-20). Next, 200 μL of 2% milk/PBS was added to each well and incubated at 37°C for 1 h. Then, 100 μL of HB2151-induced supernatant was used as a negative control; this was added to each well and incubated at 37°C for 1 h. Next, the enzyme label plate was washed three times (for 3 min each time) and the excess liquid was patted dry. Next, 100 μL (1:500) of Protein A-HRP was added to each well and incubated at 37°C for 1 h. Washing was carried out three more times (3 min each time) and excess liquid was patted dry. o-Phenylenediamine (OPD) solution (100 μL) was then added to each well and incubated at room temperature in the light for 20 min. Finally, 2 mol/L of sulfuric acid (50 μL) was added to each well to stop the reaction and the $OD_{490}$ absorption value was determined.

## Sequence determination of selected phage clones and the expression & purification of HuScFv

The M1-positive binding phage in the monoclonal ELISA was used as a template, and vector specific primers (LMB3: 5′-CAG GAA ACA GCT ATG AC-3′; PHEN: 5′-CTA TGC GGC CCC ATT CA-3′) were used to amplify the HuScFv gene fragment. The obtained amplification products were subsequently detected by 1% agarose gel electrophoresis. The PCR amplification conditions were as follows: pre-denaturation at 94°C for 5 min, 94°C for 50 s, 54°C for 50 s, 72°C for 120 s (35 cycles) and a final extension at 72°C for 10 min. The target DNA fragment was then recovered and sequenced by Kumei Biological Engineering Co. (China).

ELISA-positive strains were transferred to 5 mL of 2× TY medium containing 100 μg/mL Amp and 1% glucose and cultured overnight at 37°C. The next day, 200 μL of overnight culture was transferred to 2× TY medium (containing 100 μg/mL Amp and 0.1% glucose) and cultured at 37°C to an $OD_{600}$ of 0.9 (approximately 4 h). A final concentration of 1 mmol/L of IPTG was added for overnight induction on a shaking table at a 30°C. On the third day, the induced bacterial solution was centrifuged at $5000 \times g$ (Beckman, USA) for 30 min; the supernatant was removed and precipitated with 10%–55% saturated ammonium sulfate. The precipitated solution was then resuspended with 30 mmol/L of PB (pH7.2) and dialysis was performed in PBS overnight. The crude samples were then purified by Protein-A FF affinity chromatography; eluted samples were dialyzed with PBS overnight. The target protein was finally analyzed by 12% sodium dodecyl sulfate polyacrylamide gel electrophoresis (SDS-PAGE).

## Western blotting & immunoaffinity analysis with HuScFv

The purified M1 protein was separated by 12% SDS-PAGE and transferred to a nitrocellulose membrane using a protein electrophoresis transfer device (BIO-RAD) at 45 V for 35 min. Membranes were then probed with HuScFv as a primary antibody and Protein A-HRP as a secondary antibody (diluted with 1:2000 PBS). DAB was used as a color reagent to detect the immunobinding activity of the antibody. The binding ability of HuScFv to M1 was determined by a non-competitive ELISA method. Different concentrations of the M1 protein antigen were coated with skimmed milk powder for 1 h, and different dilutions of HuScFv were added. Then, protein A-HRP and OPD color solution were added; 2 M of sulfuric acid was used as a termination solution, and the absorbance was measured at 490 nm wavelength. Affinity

constants were calculated using the formula Ka = $(n - 1)/2(nAb' - Ab)$. IgBLAST (a tool for immunoglobulin (IG) and T cell receptor (TR) V domain sequences, NCBI) was used to analyze the nucleotide sequences of clones with the highest immune affinity constant to determine the HuScFv framework and complementary determination region (CDR).

## Expression and purification of TAT-HuScFv

Forward (5'-GTG AAT TCA TAA TGA AAT ACC TAT TGC CT-3') and reverse (5'-GCA AGC TTC TAT GCG GCC CCA TTC AG-3') sequences were introduced into *EcoR I* and *Hind III* sites, respectively. A HuScFv plasmid with a high immunoaffinity constant was extracted and used as a template for PCR. The PCR reaction conditions were as follows: pre-denaturation at 94°C for 5 min, then 34 cycles of 94°C for 50 s, 50°C for 50 s, and 72° for 10 min; this was followed by a final extension at 72°C for 10 min and cooling at 4°C. Amplification products were recovered by gel electrophoresis. The amplification products and the vector (*p*ET28a-TAT-GFP) were extracted by the double digestion of *EcoR I* and *Hind III*, respectively; the amplification product was then ligated with T4 ligase and transformed into competent *E. coli* DH5α. Positive clones were then identified by PCR.

The *p*ET28A-TAT-HuScFv construct was then transformed into chemically competent BL21(DE3) cells. A single colony was selected and inoculated into LB liquid medium (containing 50 μg/mL Kan) at 37°C and 180 rpm until the $OD_{600}$ was approximately 0.5. IPTG was added to a final concentration of 1 mmol/L and induction was carried out over a culture period of 4 h. The induced bacterial culture was centrifuged at 4°C at $5000 \times g$ for 20 min, and the bacteria were collected. The bacteria were suspended with TE (pH 8.0) buffer solution and ultrasonically crushed in an ice bath (power: 1500 W; working time: 5 s; interval time: 9 s; 50 times in total). Microscopic examination confirmed that the bacteria had been completely broken down. Centrifugation was carried out at 12,000 $g$; samples of supernatant and precipitate were then separated by 12% SDS-PAGE.

The cells expressing TAT-HuScFv and the supernatant were obtained post-lysis. The lysate was then centrifuged and purified using a metal chelated $Cu^{2+}$ column with a buffer system of PBS (pH 7.2). The target protein was then eluted with 200 mM imidazole. The eluent was then purified on a rProteinA FF column. Finally, eluted samples were dialyzed into PBS.

## Hemagglutination inhibition analysis of anti-M1-HuScFv and TAT-HuScFv

Digested MDCK cells were placed in a 96-well cell culture plate ($3\times10^4$ cells/well), and the cells grown into a single layer to be absorbed into the medium. The cells were washed three times with DMEM, and A/Changchun/01/2009 H1N1 was added to each well at different concentrations (PBS was added to the negative control well). The cells were incubated at 37°C for 3.5 h, and the extracellular fluid was discarded. Cells were washed twice with PBS; purified HuScFv and TAT-HuScFv (10 μg/well; the positive control included PBS only) were then added and incubated at 37°C for 1.5 h. The supernatant of cells was then discarded and DMEM (containing 1% FBS) was added to each well and cultured overnight at 37°C with 5% $CO_2$. The next day, the hemagglutination inhibition test was performed with the overnight culture supernatant. The supernatant was added to the reaction plate (50 μL/well), along with fresh 0.85% chicken red blood cell suspension (50 μL/well), and incubated at room temperature for 30 min; the results of the test were observed by the upright reaction plate.

**Table 1. Polypeptide sequence of the M1 protein.**

| Number | Peptide sequence based on M1 protein |
|:------:|:------------------------------------:|
| 1 | N-MSLLTEVETYVLSIIPSGPLKAEIA-C |
| 2 | N-QKLEDVFAGKNTDLEALMEWLKTRP-C |
| 3 | N-ILSPLTKGILGFVFTLTVPSERGLQ-C |
| 4 | N-RRRFVQNALNGNGDPNNMDRAVKLY-C |
| 5 | N-KKLKREITFHGAKEVALSYSTGALA-C |
| 6 | N-SCMGLIYNRMGTVTTEVAFGLVCAT-C |
| 7 | N-CEQIADSQHRSHRQMATITNPLIRH-C |
| 8 | N-ENRMVLASTTAKAMEQMAGSSEQAA-C |
| 9 | N-EAMEVANQARQMVQAMRTIGTHPNS-C |
| 10 | N-SAGLRDNLLENLQAYQKRMGVQMQRF-C |

## TAT-HuScFv bound to amino acid sites of M1 protein epitopes

The M1 protein was sequenced and then decomposed in order from the N-terminal to the C-terminal to synthesize 10 polypeptides (Table 1). Positive fragments were detected by the sandwich ELISA method, as described earlier. The polypeptides were used to coat 96-well plates, using TAT-HuScFv as the first antibody, and protein A-HRP as the second antibody. Then 3, 3′, 5, 5′- Tetramethylbenzidine (TMB) was added to visualize positive binding. Once coloration had developed, the reaction was terminated and the OD450nm was measured. The positive fragment was determined according to the value of the specific negative control. The positive fragments were decomposed into peptides according to the overlapping sequences of four amino acids and then detected by sandwich ELISA. Positive results were analyzed and positive fragments were synthesized into peptides; these were tested again by sandwich ELISA until the amino acid sites that bound to TAT-HuScFv were identified.

## Results

### Preparation of the H5N1-M1 protein

The M1 protein (H5N1, A/Meerkat/Shanghai/SH-1/2012) gene was successfully amplified and cloned into the *p*ET-Sumo-TAT vector. PCR identification revealed a target band at 750 bp (Fig 1A). Sequencing results showed that the sequence encoding the M1 protein had been cloned into the vector in the correct reading frame. The expression vector was transformed into *E. coli* BL21 (DE3) and induced with IPTG (at a final concentration of l mmol/L). The target protein (purity>95%) (Fig 1B) was successfully purified and was 30 KDa in size.

### The use of a phage display library to identify HuScFv specific to M1 H5N1

Using purified M1 protein as an antigen and the Tomlinson (I+J) phage antibody library, we identified four strains of anti-M1 protein HuScFv by four rounds of biopanning (1C, 7B, 3F, and 4G). The PCR method was used to determine whether the positive strains with biological activity had been detected successfully and specific fragments of 930 bp were amplified from the four positive strains (Fig 2A).

Western blotting was performed for the ELISA-positive strains and M1 protein. Analysis showed that 3F and 7B had strong binding ability on the M1 protein and that the stain location and stain depth were more advantageous than other forms of HuScFv under the same conditions (Fig 2B). Affinity constants for 1C, 7B, 3F, and 4G were determined by uncompetitive enzyme immunoassay, and a total of four Ka values were obtained according to the simulated

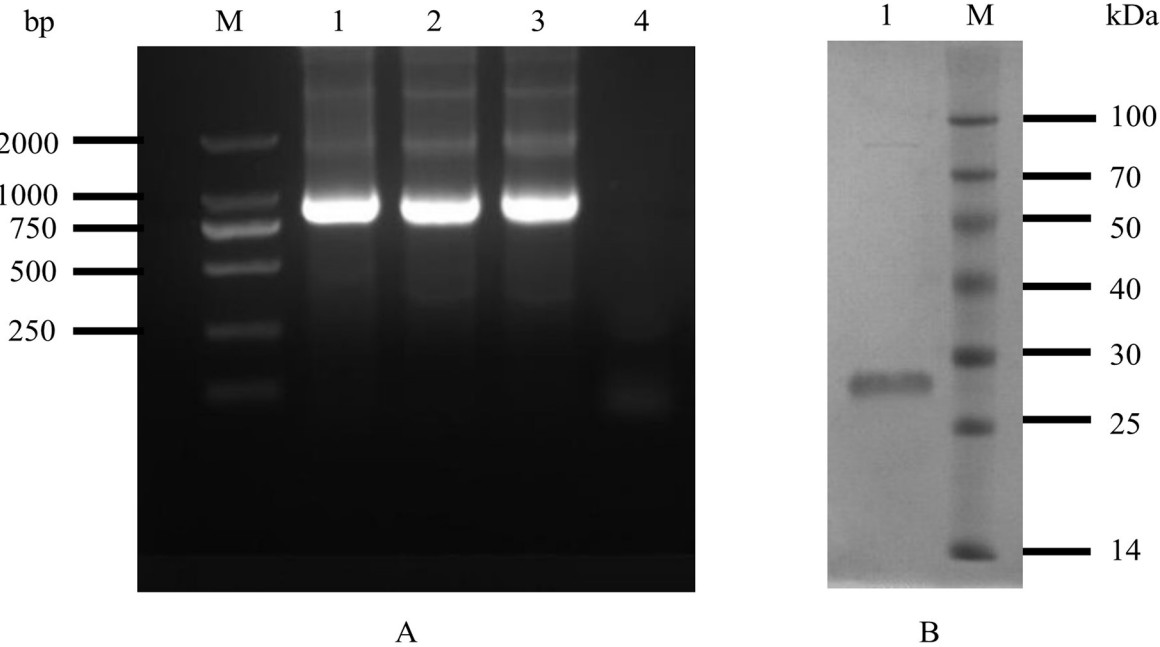

**Fig 1.** Electrophoretic images of gene amplification and PCR identification of recombinant plasmid (A) and M1 protein (B). (**A**) Lane M: DL2000; lane 1: the M1 gene was amplified with primers M1P1 and M1P2 using the H5N1 cDNA as a template; lanes 2 and 3: using the plasmid as a template, M1P1 and M1P2 amplification; lane 4, negative control; (**B**) Lane M: protein maker, lane 1: purified M1 protein.

biological curve of protein concentration for M1 (Table 2). IgBLAST was used to analyze 3F in the NCBI library to determine the HuScFv framework and CDR region (Fig 3).

## Characterization of the recombinant TAT-HuScFv trans-body for M1

The 3F and 7B genes (with the highest immune affinity constant) were successfully cloned into the *p*ET28a-TAT expression vector (Fig 4) and successfully expressed into BL21 (DE3). SDS-PAGE further showed that the protein product was approximately 28 KDa in size (Fig 5), which was the expected size of the fusion protein. TAT-HuScFv was highly expressed in IPTG-induced *E. coli*, and was eluted in 200 mmol of imidazole when purified by the metal chelated $Cu^{2+}$ column. The purified TAT-HuScFv and HuScFv were compared for hemagglutination inhibition; the ability of TAT-HuScFv when fused to the TAT domain to bind viral M1 protein was stronger than that of HuScFv (Table 3).

## Specific recognition sites of M1 protein for TAT-HuScFv

ELISA was conducted between purified TAT-HuScFv and small peptides based on M1-protein decomposition. Only fragment 10 (Table 1) was positive; all nine of the other fragments were negative. Because polypeptide number 10 had only 26 amino acid sequences; four of its amino acids were overlapped to synthesize multiple small peptides. After ELISA was conducted again, peptides 5, 6, and 7 were all positive (Fig 6A). These three small peptides all contained the same two amino acids (LENLQA, NLQAYQ, QAYQKR, QA). According to the above results, A small peptide containing two amino acids Q and A (ENLQAYQK) was selected from peptide 10, and then A peptide mutated from Q to E (ENLEAYQK) was synthesized, while another peptide mutated from A to G (ENLEQGYQK) was synthesized. Finally, ELISA (the disordered peptide with the same sequence as the source peptide was

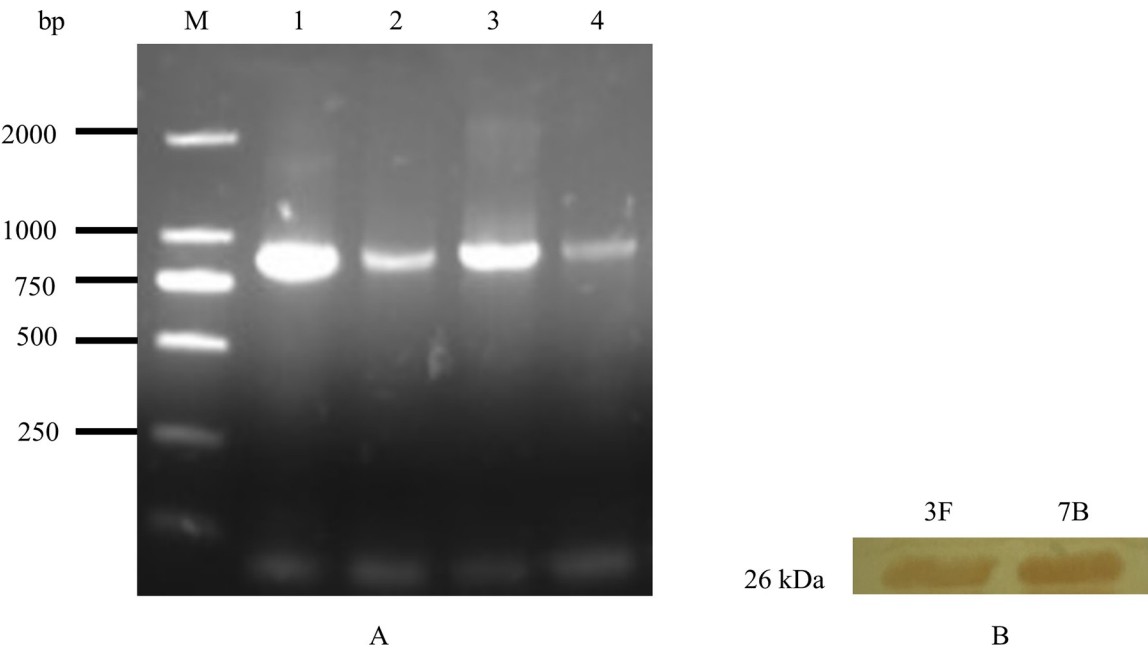

**Fig 2.** Results of PCR amplification to ELISA positive strains (A) and WB results for 3F, 7B (B). (A) Lane M: DL2000 DNA Maker. Lanes 1 and 4: specific fragments of ELISA-positive strains. (B) Under the same conditions, 3F and 7B exhibited significant advantages over other proteins.

also the negative control) was performed (Fig 6B); peptide 1 was positive and peptide 2 was negative. Therefore, it was inferred that TAT-HuScFv specifically bound to Alanine (A) at position 239 in the M1 protein.

## Discussion

New resistant strains of influenza virus are emerging all over the world and the limitations associated with existing treatment drugs for influenza virus are becoming increasingly apparent (Uyeki 2009), Consequently, there is an urgent need to develop remedies for influenza. While traditional small molecule drugs are not suitable for inhibiting the protein-protein interface (PPI), we developed a fully humanized small antibody fragment (human HuScFv) for the treatment of influenza [15, 16]. The HuScFv fragment is safe for use in different populations because it is human in origin and does not cause additional inflammation. Typically, each specific antibody molecule binds to its target using several amino acid residues in the complementary determination region (CDR) and the immunoglobulin framework (FR) of the VH and VL domains. Antibody drugs have an advantage over traditional small-molecule drugs in terms of responding to viral mutations [17]. In recent years, humanized antibodies based on HA protein encoded by influenza virus fragment 4, non-structural protein NS1

**Table 2. The affinity constants were measured as a Ka value.**

| HuScFv | n = 2 | | | n = 4 | | n = 8 | $\bar{x} \pm s$ |
|---|---|---|---|---|---|---|---|
| 1C ($\times 10^6$) | 2.7 | 3.42 | 2.95 | 3.15 | 3.09 | 3.03 | 3.06±0.22 |
| 7B ($\times 10^7$) | 1.26 | 2.4 | 4.67 | 1.85 | 8.02 | 4.55 | 3.79±2.29 |
| 3F ($\times 10^8$) | 2.8 | 2.44 | 3.95 | 2.55 | 3.27 | 3.2 | 3.04±0.51 |
| 4G ($\times 10^7$) | 3.27 | 3.97 | 2.04 | 3.71 | 2.44 | 2.53 | 2.99±0.70 |

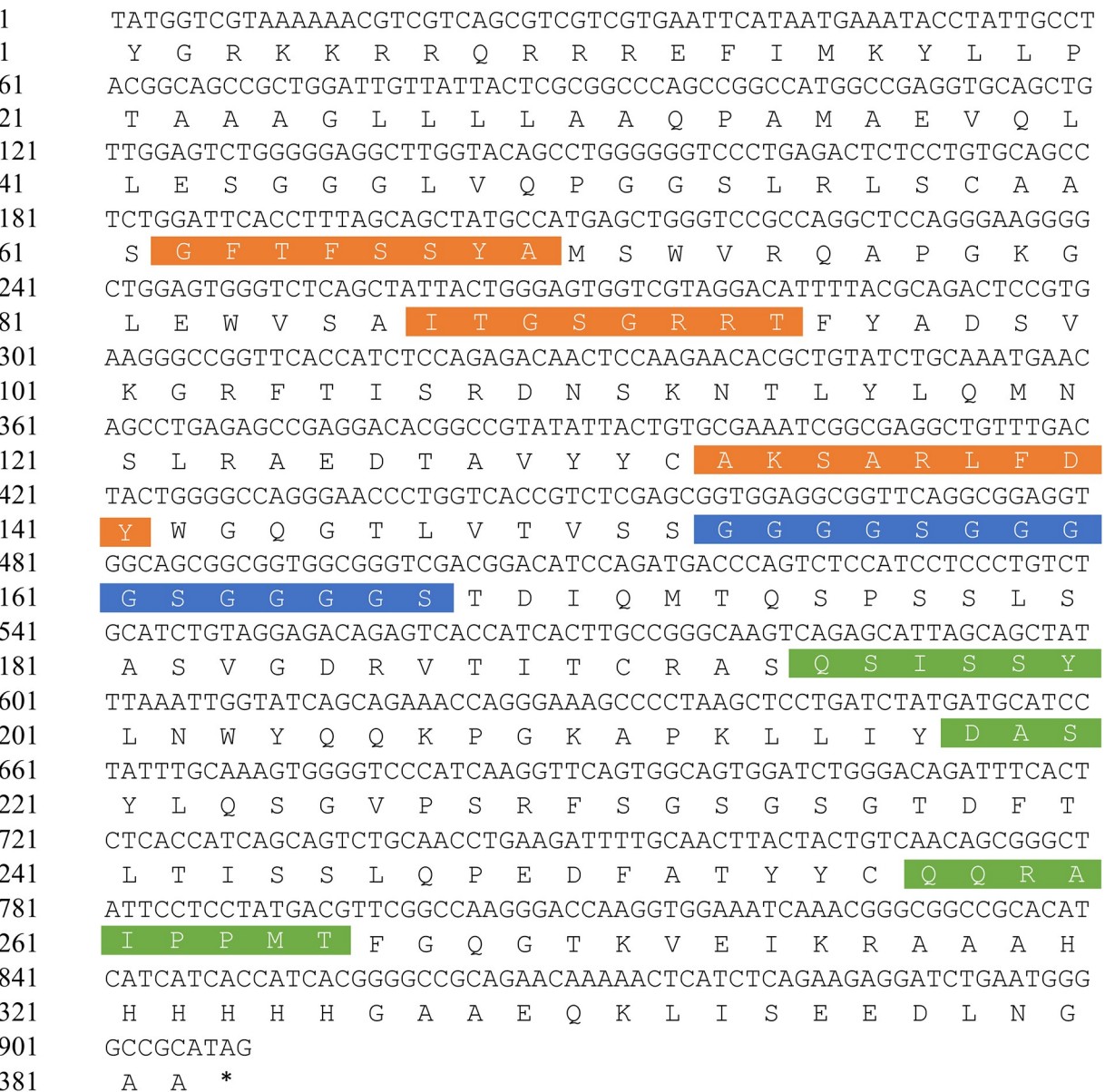

**Fig 3. DNA sequence and deduced amino acid sequence.** The orange sections show the variable regions CDR1, CDR2, and CDR3 of the heavy chain. The blue section is the protein peptide linker. The green section shows the variable regions CDR1, CDR2, and CDR3 of the light chain; the non-colored section is the frame area.

encoded by influenza virus fragment 8, and M2 protein encoded by influenza virus fragment 7, have been reported to inhibit viral activities, viral replications, and the function of viral ion channel proteins, respectively [18–22]. In addition, it has been reported that the M1 protein was highly homologous and conserved in mutant strains of influenza A [9, 10]. Consequently, specific antibodies against M1 protein will exhibit the key advantages of universality and versatility.

In the present study, we used a Tomlinson (I+J) phage antibody library to isolate a unique HuScFv that exhibited high affinity for the influenza virus M1 protein. By sequencing the HuScFv gene, we were able to fuse an HIV TAT protein transduction domain to the HuScFv

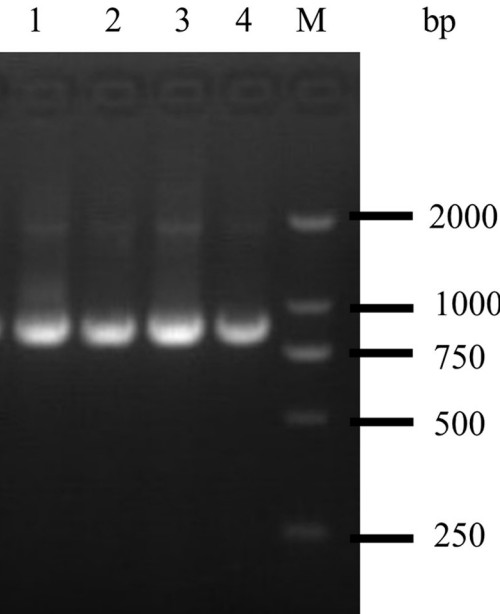

**Fig 4. PCR identification of the recombinant expression plasmid pET28a-TAT-HuScFv.** Lane M: DL2000 DNA marker; lanes 1 and 4: PCR product for the recombinant expression plasmid *p*ET-28a-TAT-HuScFv.

fragment. In the hemagglutination inhibition experiment using MDCK cells, we demonstrated that TAT-HuScFv could penetrate the cell membrane more quickly than traditional HuScFv and then penetrate cells and bind specifically to M1 protein under the same conditions. We also revealed that the linkage of TAT-HuScFv to TAT is more effective, faster and stronger than traditional HuScFv in terms of practical application. In subsequent experiments, we will conduct large-scale animal model experiments to clearly demonstrate the differences between TAT-HuScFv and HuScFv. The binding site was proven to be the Alanine (A) at position 239

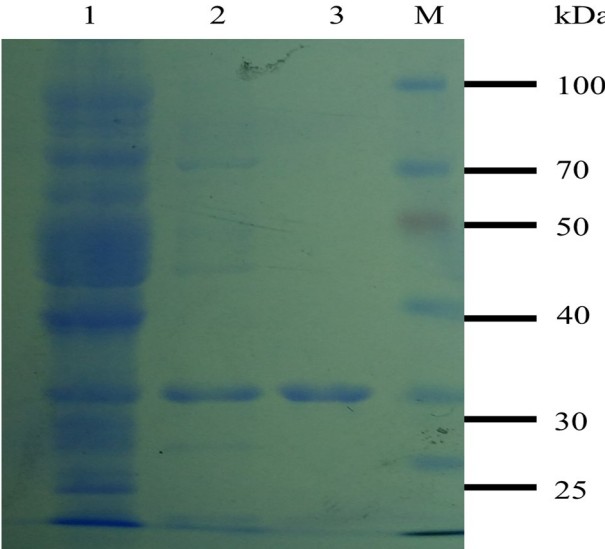

**Fig 5. Purified TAT-HuScFv.** Lane M: marker; lane 1: ultrasonic supernatant induced by TAT-HuScFv expression bacteria; lane 2: Cu$^{2+}$ 200 mmol imidazole eluted sample; lane 3: purified TAT-HuScFv.

**Table 3. Hemagglutination inhibition test results.**

| Antibody | H1N1* (TCID$_{50}$) | | | | | | | |
|---|---|---|---|---|---|---|---|---|
| | **50** | **100** | **150** | **200** | **250** | **300** | **350** | **400** |
| 3F | - | - | - | - | + | + | + | + |
| TAT-3F | - | - | - | - | - | - | + | + |
| 7B | - | - | - | - | + | + | + | + |
| TAT-7B | - | - | - | - | - | + | + | + |
| Positive control | + | + | + | + | + | + | + | + |
| Negative control | - | - | - | - | - | - | - | - |

*The TCID$_{50}$ of H1N1 was $10^{-4.5}$/0.1mL, positive (+), negative (-).

in the M1 protein. When used to treat influenza virus infection, the penetrating TAT-HuScFv could significantly shorten the treatment time and improve the efficiency of treatment. In conclusion, the penetrating TAT-HuScFv generated in this study could substantially neutralize a large number of viruses in mammalian cells, and exerted significant effects on the life cycles of viruses. Therefore, this study provides a solid foundation for the protection and treatment of avian influenza virus H5N1 and its related subtypes.

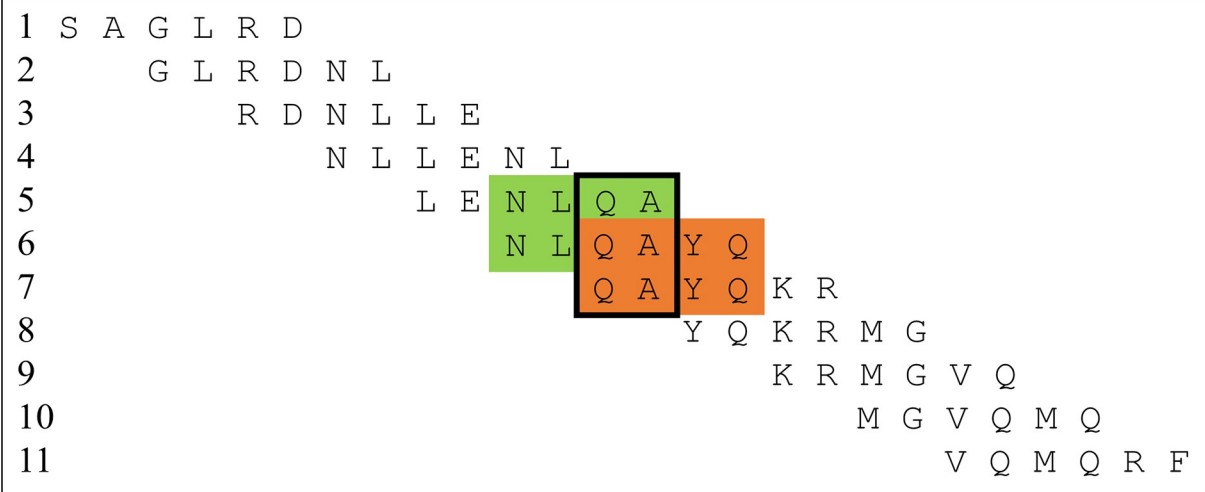

A

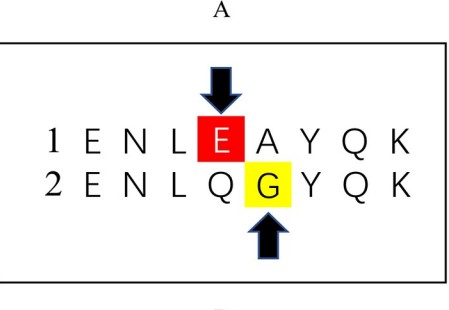

B

**Fig 6.** Schematic of peptide fragment 10 decomposed into small peptides (A) and amino acid mutation in polypeptides (B). (A): The green and orange sections are ELISA-positive, and the black circled sections are the amino acid sequences shared by the three polypeptides 5, 6 and 7. (B): The red and yellow sections represent the mutant amino acid sites that are positive for fragment 1 and negative for fragment 2.

## Conclusions

We developed a recombinant HuScFv antibody with high affinity for influenza virus M1 protein by using the Tomlinson (I+J) phage antibody library. TAT-HuScfv was further enhanced by binding to HuScFv fragment via the TAT protein transductive domain of the HIV virus. The binding site was identified as alanine (A) at the 239th position of the M1 protein.

## Supporting information

**S1 File.**
(ZIP)

## Acknowledgments

The authors would like to thank Dr. Guoli Zhang for data analysis and the support of the Changchun Veterinary Research Institute, Chinese Academy of Agricultural Sciences. We would also like to thank International Science Editing (http://www.internationalsciencee diting.com) for editing this manuscript.

## Author Contributions

**Conceptualization:** He Sun, Guoli Zhang, Zehong Li.

**Data curation:** He Sun, Yu Wang, Hongyang Man, Guoli Zhang.

**Formal analysis:** He Sun, Hongyang Man, Zehong Li.

**Funding acquisition:** Guoli Zhang.

**Investigation:** Zehong Li.

**Methodology:** Guangmou Wu, Jiyuan Zhang, Yu Wang.

**Project administration:** He Sun, Guoli Zhang.

**Resources:** He Sun.

**Software:** He Sun, Jiyuan Zhang, Yue Qiu.

**Supervision:** Guangmou Wu, Yuan Tian.

**Validation:** Yue Qiu, Yuan Tian.

**Visualization:** Guoli Zhang.

**Writing – original draft:** He Sun.

**Writing – review & editing:** Guoli Zhang, Yuhuan Yue, Yuan Tian.

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
