## [Decision Letter · Decision Letter 0]

16 Dec 2021

PONE-D-21-35307Characterization of An Intracellular Humanized Single-Chain Antibody to Matrix Protein (M1) of H5N1 VirusPLOS ONE

Dear Dr. Yuhuan,

Thank you for submitting your manuscript to PLOS ONE. After careful consideration, we feel that it has merit but does not fully meet PLOS ONE’s publication criteria as it currently stands. Therefore, we invite you to submit a revised version of the manuscript that addresses the points raised during the review process.

We look forward to receiving your revised manuscript.

Kind regards,

Hari S. Misra

Academic Editor

PLOS ONE

Journal Requirements:

Additional Editor Comments:

Dear Dr Yue Yuhuan,

Thank you for submitting your work to PLoS ONE. This has been reviewed by 2 experts in the field. Both of them have appreciated it but have also suggested some work for further improvement. I suggest to revise manuscript it by addressing all the concerns of both the reviewers.

Reviewers' comments:

Reviewer's Responses to Questions

**Comments to the Author**

1. Is the manuscript technically sound, and do the data support the conclusions?

Reviewer #1: Partly

Reviewer #2: Yes

2. Has the statistical analysis been performed appropriately and rigorously? 

Reviewer #1: I Don't Know

Reviewer #2: N/A

3. Have the authors made all data underlying the findings in their manuscript fully available?

Reviewer #1: No

Reviewer #2: Yes

4. Is the manuscript presented in an intelligible fashion and written in standard English?

Reviewer #1: Yes

Reviewer #2: Yes

5. Review Comments to the Author

Reviewer #1: This study by He Sun et. al. reports a novel single chain humanized antibody fragment against the M1 protein of the avian influenza virus generated using phage library. The methodology is well described, and the results are well summarized. The manuscript needs further improvement including sharing of raw data for the results presented and an overall copy editing. Refer to the queries listed below.

Major comments:

• This study reports multiple ELISA assays, however the results of these assays (images of the plates at the least or the actual read outs) have not been provided, the absence of which makes the study look superficial and non-authentic.

• Similarly for the statement “The purified TAT-HuScFv and HuScFv were compared for hemagglutination inhibition; the ability of TATHuScFv when fused to the TAT domain to bind viral M1 protein was stronger than that of HuScFv (Table 3)”, no actual data has been presented in the manuscript, making the interpretations vague.

• The authors need to examine the efficacy of the TAT-HuScFv and HuScFv fragments in neutralizing/inhibiting viral infection in a suitable ex vivo infection model.

Minor Comments:

• Rephrase the sentence “The 16 sub-types of hemagglutinin HA (H1-H16) and the nine sub-types of neuraminidase NA (N1-N9) are proteins involved in avian influenza” to express it more clearly.

• Provide a reference for the statement “The pET28a-TAT-GFP vector was constructed previously by our research group.”

• What was the nature of competency in the BL 21 cells mentioned in the sentence “The vector was sequenced and then transformed into BL21 (DE3) competent cells.”

• Replace “eluded” with “eluted” in the phrase “The eluded phage was cloned into E. coli TG1”.

• Rephrase the sentence “The pET28A-TAT-HuScFv construct was then transformed into BL21(DE3) with the CaCl2 method.” to “The pET28A-TAT-HuScFv construct was then transformed into chemically competent BL21(DE3) cells.”

• Rephrase this sentence “The supernatant of the expressed cells was obtained following ultrasonic lysis”. I believe the authors intend to say cells expressing the TAT-HuScFv and the supernatant was obtained post lysis and centrifugation of the lysate.

• Rephrase “The supernatant of cells was then absorbed”.

• Rephrase and expand to make this statement clear “Analysis involved 3, 3′,5 ,5′- Tetramethylbenzidine (TMB) color and the positive fragments.”

• Rephrase “this was the expected size of the fusion protein” as “which was the expected size of the fusion protein”.

• Rephrase with a clearer description “Two polypeptides, containing eight amino acids (QA) were synthesized; then, Q A was genetic mutated into E (ENLEAYQK) G(ENLEQGYQK) respectively.”

• Penetrate instead of “penetrat” in the sentence “we demonstrated that TAT-HuScFv can quickly penetrat the cell”.

• The main text includes a lot of complex sentences (often joined using semi-colon), making it hard to read. I suggest the author perform a careful copy-editing of the manuscript and reduce such sentences.

Reviewer #2: Author Sun et. al., provides well characterized humanized antibodies against M1 protein, which showed excellent anti-influenza activity as shown by hemagglutination assay and found out that the single amino acid change may alter the binding. Here are my comments:

Abstract section:

1- Change the sentence to “cDNA of the H5N1 virus as a template; the M1 protein was then expressed and purified” to ‘cDNA of the H5N1 virus as a template, expressed in bacterial expression system (name the bacteria used) and purified.’

2- Space in 300 TCID50 line no 09; against the; line no 12.

3- Please use correct grammar and sentences in writing.

Method section:

1- First para line-3, provide the reference.

2- If virus was used in the study, please provide, the preparation method, stock concentration estimation etc.

3- Whether PCR amplicon of M1 was PCR/gel purified and any restriction enzymes were used to ligate into bacterial vector.

4- Please provide purification method for M1 protein

5- Please re-write this long sentence ‘Next, the Tomlison I+J phage antibody library was added and diluted with 2% milk/PBS to a titer of 1.0 ×1013; 100 μL was added to each well, and the liquid was incubated with vigorous shaking at room temperature for 60 min.’

6- WB analysis paragraph, used probed instead of ‘blotted’

7- Add extension time of PCR in Expression and Purification of TAT-HuScFv paragraph

8- First line of Hemagglutination Inhibition Analysis of Anti-M1-HuScFv and TAT-HuScFv ‘Digested MDCK cells’ with which enzyme?

9- Line 1-2; page no 12; please highlight or bold the changed amino acid in the small peptides.

10- Provide the role of GFP in this construct pET 28 TAT- HuScFv.

Results section:

Please provide a full blot of figure-2B.

In discussion, please provide the comparison of anti M1-TAT-HuScFv and previously published humanized antibodies against M1 protein.

6. PLOS authors have the option to publish the peer review history of their article (what does this mean?). If published, this will include your full peer review and any attached files.

Reviewer #1: No

Reviewer #2: **Yes: **RAVI P. ARYA

---

## [Author Response · Author response to Decision Letter 0]

3 Mar 2022

Dear editor:

 Thank you very much for your comments and I will reply one by one here.

I understand.

We received two grants, Jilin Province Science and technology development plan.

20200404113YY, 20190304038YY, Please find the attached file A.

No changes need to be made.

I understand.

I understand.

I understand. After my careful review, there are no retracted articles.

Reviewer #1:

1. This study reports multiple ELISA assays, however the results of these assays (images of the plates at the least or the actual read outs) have not been provided, the absence of which makes the study look superficial and non-authentic.

We can provide real data for your review. Please find the attached file B.

2. Similarly for the statement “The purified TAT-HuScFv and HuScFv were compared for hemagglutination inhibition; the ability of TATHuScFv when fused to the TAT domain to bind viral M1 protein was stronger than that of HuScFv (Table 3)”, no actual data has been presented in the manuscript, making the interpretations vague.

It can be clearly seen from Table 3 that in the comparison of hemagglutination inhibition between TAT-HuScFv and HuScFv, the results of hemagglutination of TAT-3F and TAT-7B are better than those of 3F and 7B.

3. The authors need to examine the efficacy of the TAT-HuScFv and HuScFv fragments in neutralizing/inhibiting viral infection in a suitable ex vivo infection model.

What you said is very correct and agrees with your point of view. This is exactly what our research group is doing now, and we will work hard to improve it. My classmate is working on the titer of TAT-HuScFv in the mouse model.

4. Rephrase the sentence “The 16 sub-types of hemagglutinin HA (H1-H16) and the nine sub-types of neuraminidase NA (N1-N9) are proteins involved in avian influenza” to express it more clearly.

I understand.

5. Provide a reference for the statement “The pET28a-TAT-GFP vector was constructed previously by our research group.”

This vector was constructed by the senior sister Xu Yanling from the previous research group and published in the Chinese core journal "Chinese Journal of Biological Products"-Purification of PTD-GFP fusion protein and determination of its transduction efficiency. Please find the attached file C.

6. What was the nature of competency in the BL 21 cells mentioned in the sentence “The vector was sequenced and then transformed into BL21 (DE3) competent cells.”

BL21 (DE3) competent cells were obtained by using escherichia coli BL21 (DE3) strain through special processing, which can be used for chemical transformation of DNA. BL21 (DE3) strain was suitable for expressing non-toxic proteins, and the strain was the host of high expression of exogenous gene proteins with T7 RNA polymerase as expression system. Using pU19 plasmid detection, the conversion efficiency can reach 107, and the conversion efficiency does not change when stored at -80℃ for a long time.

7. Replace “eluded” with “eluted” in the phrase “The eluded phage was cloned into E. coli TG1”.

I understand.

8. Rephrase the sentence “The pET28A-TAT-HuScFv construct was then transformed into BL21(DE3) with the CaCl2 method.” to “The pET28A-TAT-HuScFv construct was then transformed into chemically competent BL21(DE3) cells.”

I understand.

9. Rephrase this sentence “The supernatant of the expressed cells was obtained following ultrasonic lysis”. I believe the authors intend to say cells expressing the TAT-HuScFv and the supernatant was obtained post lysis and centrifugation of the lysate.

I understand.

10. Rephrase “The supernatant of cells was then absorbed”.

I understand.

11. Rephrase and expand to make this statement clear “Analysis involved 3, 3′,5 ,5′- Tetramethylbenzidine (TMB) color and the positive fragments.”

I understand.

12. Rephrase “this was the expected size of the fusion protein” as “which was the expected size of the fusion protein”.

I understand.

13. Rephrase with a clearer description “Two polypeptides, containing eight amino acids (QA) were synthesized; then, Q A was genetic mutated into E (ENLEAYQK) G(ENLEQGYQK) respectively.”

I understand.

14. Penetrate instead of “penetrat” in the sentence “we demonstrated that TAT-HuScFv can quickly penetrat the cell”.

I understand.

15. The main text includes a lot of complex sentences (often joined using semi-colon), making it hard to read. I suggest the author perform a careful copy-editing of the manuscript and reduce such sentences.

I understand.

Reviewer #2:

1. Change the sentence to “cDNA of the H5N1 virus as a template; the M1 protein was then expressed and purified” to ‘cDNA of the H5N1 virus as a template, expressed in bacterial expression system (name the bacteria used) and purified.’

I understand.

2. Space in 300 TCID50 line no 09; against the; line no 12.

I understand.

3. Please use correct grammar and sentences in writing.

This manuscript has been edited by a native English professor with a doctorate degree and the International Science Editing corporation again.

4. First para line-3, provide the reference.

I understand.

5. If virus was used in the study, please provide, the preparation method, stock concentration estimation etc.

The H1N1 virus was passed by MDCK cells. After the MDCK cells were covered with a single layer, 2 mL DMEM medium containing H1N1 was added into 25 cm2 culture flask for incubation for 1 h, and 3 mL DMEM (containing 0.5% TPCK trypsin) was added, and the virus was recovered after 48 h. The virus TCID50 was 105.37 by Reed-Muench method.

6. Whether PCR amplicon of M1 was PCR/gel purified and any restriction enzymes were used to ligate into bacterial vector.

The PCR amplicon of M1 was PCR/gel purified. And any restriction enzymes weren't used to ligate into bacterial vector.

7. Please provide purification method for M1 protein.

I understand.

8. Please re-write this long sentence ‘Next, the Tomlison I+J phage antibody library was added and diluted with 2% milk/PBS to a titer of 1.0 ×1013; 100 μL was added to each well, and the liquid was incubated with vigorous shaking at room temperature for 60 min.’

I understand.

9. WB analysis paragraph, used probed instead of ‘blotted’.

I understand.

10. Add extension time of PCR in Expression and Purification of TAT-HuScFv paragraph.

I understand.

11. First line of Hemagglutination Inhibition Analysis of Anti-M1-HuScFv and TAT-HuScFv ‘Digested MDCK cells’ with which enzyme?

Trypsin commonly used to digest cells.

12. Line 1-2; page no 12; please highlight or bold the changed amino acid in the small peptides.

I understand.

13. Provide the role of GFP in this construct pET 28 TAT- HuScFv.

The GFP tag has no effect, because the corresponding vector used is left over from a previous laboratory, and although its name can be seen in this study, it will not play any role.

14. Please provide a full blot of figure-2B.

I understand. Please find the attached file D.

15. In discussion, please provide the comparison of anti M1-TAT-HuScFv and previously published humanized antibodies against M1 protein.

I understand.

---

## [Editor Report · Decision Letter 1]

17 Mar 2022

Characterization of an Intracellular Humanized Single-Chain Antibody to Matrix Protein (M1) of H5N1 Virus

PONE-D-21-35307R1

Dear Dr. Yuhuan,

We’re pleased to inform you that your manuscript has been judged scientifically suitable for publication and will be formally accepted for publication once it meets all outstanding technical requirements.

Kind regards,

Hari S. Misra

Academic Editor

PLOS ONE
---

## [Editor Report · Acceptance letter]

22 Mar 2022

PONE-D-21-35307R1 

Characterization of an Intracellular Humanized Single-Chain Antibody to Matrix Protein (M1) of H5N1 Virus 

Dear Dr. Yue:

I'm pleased to inform you that your manuscript has been deemed suitable for publication in PLOS ONE. Congratulations! Your manuscript is now with our production department. 

Kind regards, 

on behalf of

Professor Hari S. Misra 

Academic Editor

PLOS ONE